# ViTFER: Facial Emotion Recognition with Vision Transformers

**Aayushi Chaudhari [1], Chintan Bhatt [2],\*, Achyut Krishna [1] and Pier Luigi Mazzeo [3],\***

[1]   U & P U. Patel Department of Computer Engineering, Chandubhai S Patel Institute of Technology (CSPIT), CHARUSAT Campus, Charotar University of Science and Technology (CHARUSAT), Changa 388421, India
[2]   Department of Computer Science and Engineering, School of Technology, Pandit Deendayal Energy University, Gandhinagar 382007, India
[3]   Institute of Applied Sciences and Intelligent Systems, National Research Council of Italy, 73100 Lecce, Italy
\*   Correspondence: chintan.bhatt@sot.pdpu.ac.in (C.B.); pierluigi.mazzeo@cnr.it (P.L.M.)

**Abstract:** In several fields nowadays, automated emotion recognition has been shown to be a highly powerful tool. Mapping different facial expressions to their respective emotional states is the main objective of facial emotion recognition (FER). In this study, facial expression recognition (FER) was classified using the ResNet-18 model and transformers. This study examines the performance of the Vision Transformer in this task and contrasts our model with cutting-edge models on hybrid datasets. The pipeline and associated procedures for face detection, cropping, and feature extraction using the most recent deep learning model, fine-tuned transformer, are described in this study. The experimental findings demonstrate that our proposed emotion recognition system is capable of being successfully used in practical settings.

**Keywords:** computer vision; emotion recognition; ResNet; transformers; Vision Transformers

## 1. Introduction

Emotion recognition helps humans understand the intentions and psychic conditions of the human mind. Nevertheless, an individual will not be able to recognize the human mentality accurately. Emotions are fundamental features of humans that play an important role in social communication [1,2]. Automated emotion recognition can help numerous domains such as education, healthcare, cybersecurity, human–computer interaction (HCI), human–robot interaction (HRI), and virtual reality to sense the stimuli of the human brain by applying various modalities such as images, videos, audio, and text. Humans express emotion in various ways, including facial expression [3,4], speech [5], and body language [6]. Facial expression analysis is the most popular and well-researched element related to emotion recognition. The main objective of this research is to investigate and check the feasibility of the Vision Transformer in the emotion recognition domain by utilizing a combination of well-known datasets for FER. High-quality data are being gathered in this area to meet the system's requirements for sensing, processing, and simulating human emotions.

The portrayal of the six fundamental human emotions—happiness, anger, surprise, sadness, fear, and disgust—by humans is a well-established fact [7]. These are the six basic emotions, other than these, several other pieces of research are considered for research according to the respective domain. They include various emotions such as drowsiness, pain, and tiredness. Various areas offer the potential for the application of automated facial expression recognition. Vital security information can be retrieved from live monitoring of people's facial expressions. An alert can be generated from a computer system when a human exhibits fear, disgust, pain, or any such kind of facial expression; this can help medical professionals to keep track of a patient's mental state. When something is found awry, professionals can change their method of treating the patient. Various machine learning and deep learning algorithms are being applied along with some novel approaches

to obtain effective results for emotion recognition. In this research, we will focus on the latest deep learning model, i.e., a transformer with an attention mechanism. The pivotal mode of recognizing human emotion is thought to be facial expression. This mode uses facial motion and curvatures of facial characteristics to classify emotions into various categories. Making emotion recognition systems functional for real-life circumstances is one of our priorities. Many researchers in psychology have found that participants correctly identified the emotions represented by the body features when faces and body traits representing various emotions were combined. Faces are one of these features that are frequently used, if not the most frequently used, for various reasons, including their visibility, the wealth of features they contain for emotion recognition, and the ease with which a large dataset of faces can be gathered. Here, we consider facial expressions in the form of images as an input modality for recognizing the emotional state of a human mind. It is interesting that, in the case of facial expressions, the lips and eyes provide most cues, whereas other facial features, such as the ears and hair, play a small role [8]. Considering the facial expression as the relevant cue for the emotion recognition system, we also represented a new dataset named AVFER, which is a combination of FER2013, AffectNet, and CK+48 datasets. We applied fine-tuned ViT with a FER-based base configuration for image recognition and test its robustness on AVFER. Firstly, we analyzed various datasets for facial emotion recognition by identifying the distribution ratio of emotion categories. After rectifying the uneven distribution of data, we decided to merge three datasets, namely FER-2013, AffectNet, and CK+48 in the fraction of proportions. We then split a set of sample data into training, validation, and testing to evaluate our model. We applied fine-tuned ViT with a FER-based base configuration for image recognition and tested its robustness on AVFER. Our aim was to test and develop a model that could identify eight classes of emotions. Additionally, for the following model testing and evaluation, we contrasted the ViT-B/16 fine-tuned model with the similarly fine-tuned ResNet model. We then decided to analyze the results by applying AVFER on the following models: ResNet-18, Vit-b/16/s, ViT-B/16/SG, and ViT-B/16/SAM.

## 2. Related Work

Several facial emotion recognition approaches have been experimented with over the previous few decades. Traditional pioneer approaches retrieve features from a facial image and then classify emotion based on feature values. Recent deep learning-based algorithms, on the other hand, work out the emotion recognition problem by integrating both processes into a single composite operating process. Several papers have examined and compared existing FER approaches [9–12] with the most recent [11,12] deep learning-based methods. In this paper, our aim is to examine the most recent deep learning model, called Vision Transformer, for recognizing facial emotions by taking an image of a facial expression as an input parameter. The popular approaches used in the emotion recognition methods are briefly described in the following subsections.

### 2.1. Emotion Recognition Methodologies Based on Machine Learning

Various classic machine learning approaches such as K-nearest neighbor and support vector machines are used to address the facial emotion recognition challenge. Feng et al. [13] generated local binary pattern (LBP) histograms from multiple small segments of the image, integrated them into a single feature histogram, and then classified emotion using a linear programming (LP) technique. Xiao-Xu and Wei [14] first added a wavelet energy feature (WEF) to a face picture, extracted features using Fisher's linear discriminant (FLD), and then classified emotion using the K-nearest neighbor (KNN) method. Lee et al. [15] implemented a boosting technique for classification and an enhanced wavelet transform for 2D feature extraction, termed contourlet transform (CT). Chang and Huang [16] integrated face recognition into FER to improve individual expression recognition, and they employed a radial basis function (RBF) neural network for classification.

Alshamsi et al. [17] used SVM to analyze two feature descriptors, facial landmarks descriptor and center of gravity descriptor, and applied them for facial feature extraction. In their thorough study, Shan et al. [18] assessed several facial renderings depending on local statistical features and LBPs using different forms of SVM. The primary downside of the observed conventional approaches is that they only evaluated frontal images for facial emotion recognition since features from front-face views differ in typical feature extraction methods.

### 2.2. Emotion Recognition Methodologies Based on Deep Learning

Deep learning in the emotion recognition domain is a relatively well-known paradigm in machine learning, with some CNN-based research already described in the literature. A study by Pranav et al. [19] investigated FER on self-collected face emotional images using a typical CNN architecture with two convolutional-pooling layers. Pons et al. [20] developed an ensemble of 72 CNNs by training individual CNNs with varied filter sizes in convolutional layers or different numbers of neurons.

Ding et al. [21] have presented an architecture that integrates a deep facial recognition framework with the FER. The FaceNet2ExpNet extension of FaceNet, which was already being used by Google's artificial intelligence (AI) team, was improved by Li et al. [22] via transfer learning. An affective computing system that recognises emotion using music, video, and facial expressions was demonstrated by Yagyaraj et al. [23]. For reducing the computation cost of the neural networks, the standard 2D/3D convolution layers are factorized into separate channels and spatiotemporal interactions. Amir et al. [24] proposed the Learnable Graph Inception Network (L-GrIN), which simultaneously learns to perceive emotion and identify the underlying graph structure in the dynamic data. The authors used three different modalities (video, audio, and motion capture) in the form of facial expressions, voice, and body motions, respectively. Ronak et al. [25] presented Facial Affect Analysis Based on Graphs, which combines two alternative methods for representing emotions: (1) a set of 26 discrete categories and (2) the continuous dimensions of valence, arousal, and dominance, using various CNN models for recognizing emotions. Soumya et al. [26] developed a multimodal conversational emotion recognition system employing speech and text; the authors used a bidirectional GRU network with self-attention to separately process textual and auditory representations. The Attention-Based Magnification-Adaptive Network is a network that Mengting et al. [27] proposed that learns adaptive magnification levels for the microexpression (ME) representation (AMAN). The network is made up of two modules: the frame attention (FA) module, which focuses on discriminative aggregated frames in a ME video, and the magnification attention (MA) module, which automatically focuses on the proper magnification levels of different MEs. Thus, we can see that various studies have been conducted on FER approaches. In this research paper, we will represent various models of transformers, which are the latest deep learning model.

The literature study mentioned above illustrates the various machine learning and deep learning techniques used for emotion classification. We acknowledge the standard methods for facial emotion recognition that are definitely showing promising results.

### 3. Data Integration/Augmentation and Analysis

This section discusses the features of each dataset described below and lists any characteristics useful for data integration on the AVFER dataset, which is the merged dataset, as transformers need a good amount of samples to retrieve hidden patterns during the training phase and the few data in our hands are not enough to satisfy this requirement. Therefore, we have the plan to manipulate our small amount of samples to increase the size of the final datasets using data augmentation; the final dataset will have eight different classes integrated by three different subsets. We have only used samples available on Kaggle or other open-source data platforms for preprocessing. In data analysis, we used

some Python libraries to display and list images or their features such as *glob, matplotlib, cv2, imageio, PIL, NumPy, pandas, and matplotlib.*

*Datasets*

**FER-2013:** FER-2013 is a dataset composed of 35.953 images in seven classes (fear, disgust, sad, happy, neutral, surprise, angry). As seen in Figure 1, Images are 48 × 48 pixels in size with a grey-scaled color palette. The classes' variations and feature distributions are helpful in the merging phase for other classes to obtain a good distribution and normalize the amount of data variation. According to the final classification, the contempt class was missed for this type of data. Its composition already divides its original samples into training and validation sets using different folders with the function of labeling.

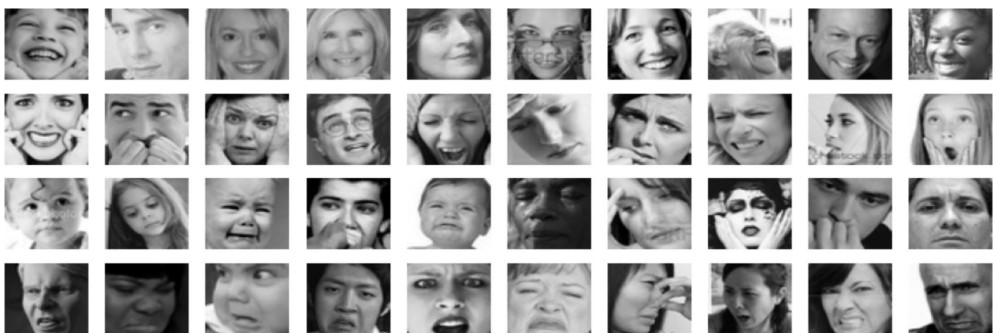

**Figure 1.** Samples from the FER-2013 dataset.

We cannot use this structure because it is necessary to merge it with other datasets and proceed with the splitting into three different subfolders (training, testing, and validation). Finally, we can analyze some features of the sample distribution separately between training and validation folders. Figure 2 represents the graph distribution of various classes in the FER2013 dataset, where red bars indicate values below the mean and green indicates values above the mean, where means are equal to 4100 and 1100 for training and validation sets, respectively.

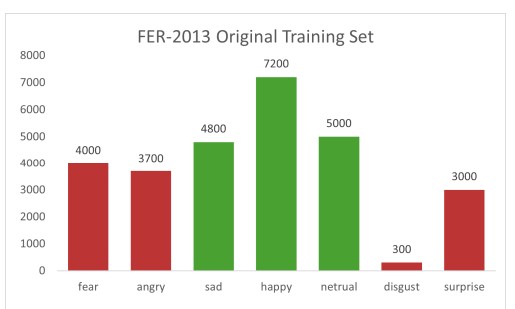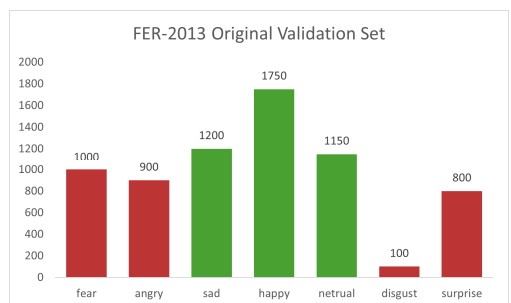

**Figure 2.** Distribution of classes in the FER-2013 dataset. Graph represents the mean values of green and red bars.

FER-2013 does not have many samples for the disgust class. This observation provides excellent motivation to merge it with other datasets and perform the data augmentation phase. Furthermore, it is unbalanced, and we must manage it in data augmentation proportionally or data integration with some samples from other datasets to obtain a result set with the same number of samples for each class.

- **CK+48:** CK+48 is a small dataset composed of 981 images in seven classes (fear, disgust, sad, happy, neutral, surprise, angry). Images are 48 × 48 in size with a grey-scaled color palette as shown in Figure 3. The classes' variations and feature distributions are helpful in the merging phase for other classes to obtain a good distribution and

normalize the amount of data variation. Generally, images taken from video frames did not have much variation, and the total number of elements is negligible compared to other datasets. Compared with the FER-2013, images are in frontal view with a clean pattern for facial expression.

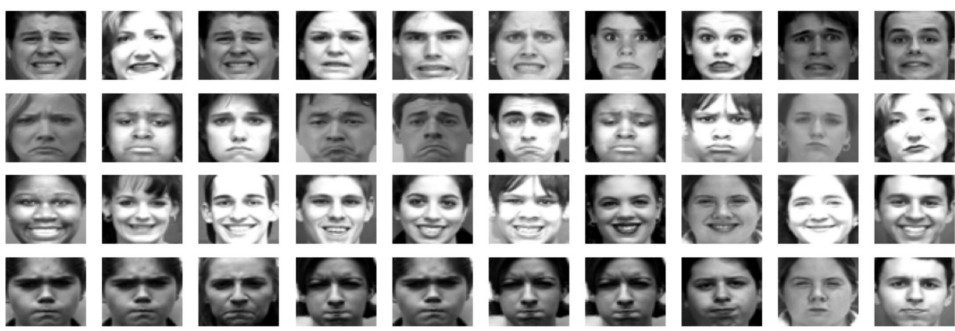

**Figure 3.** Samples from the CK+48 dataset.

CK+48 contained less than 1000 images and did not have a splitting into the training and validation sets as FER-2013. We will use it as part of the final training set for each dataset integration in our experiments.

Figure 4 represents the graph distribution of various classes in the CK+48 dataset, where red bars indicate values below the mean and green indicates values above the mean, where the mean is equal to 142. According to the final classification, we do not have any samples for the disgust class, and contempt has fewer samples; pixel values are in the total range (0.255) for a single channel. Images are in PNG format; we will convert them to JPG to normalize images under a single format. The algorithms used for compression are the primary distinction between JPG and PNG. To make the file smaller, JPG employs a lossy compression method that discards some of the image data. PNG, in contrast, employs a lossless technique that preserves all the data; however, given the great number of files that we will generate from data augmentation, we decided to have lower-quality resolution images with reduced size compared to high-quality ones because of the small amount of data in PNG; the image noise applied on their conversion did not have a proportionally huge impact on the final dataset, and some added noise can increase the resistance of the model.

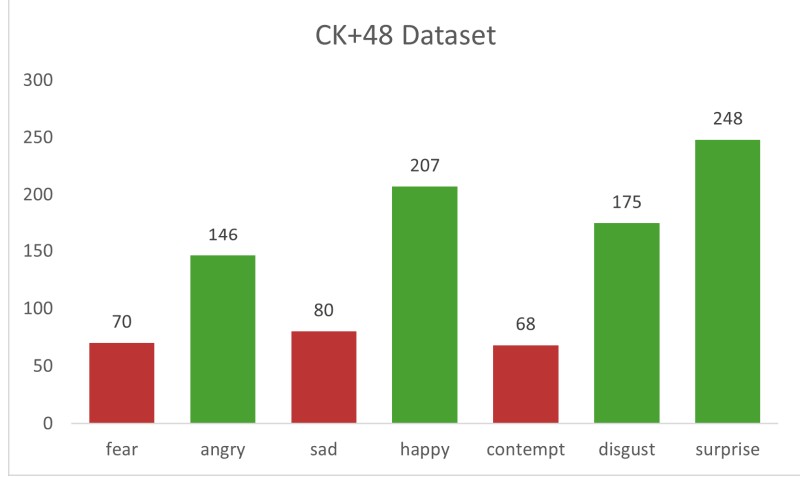

**Figure 4.** Distribution of the CK+48 dataset with the classification of classes describing their mean values.

- **AffectNet:** The AffectNet dataset [28] has samples of different sizes and high-quality images in gray-scale or color in the RGB range. It has eight different classes (surprise, angry, sad, contempt, disgust, fear, neutral, and happy). As of the FER-2013, there is a division between the validation and training sets, but subfolders are almost balanced. We will exploit this dataset for the training, testing, and validation splitting, given the validation set in the same subfolder for the final set, and balance original training samples between the final testing and training subset for some experiments. In the last experiments, with a hybrid validation set, we will use the original AffectNet validation set in the testing phase.

As can be seen from Figure 5, AffectNet has some images in color and some in black and white with different face angles and backgrounds. During the data analysis, we ensured the correct channel distribution for the entire original dataset. In other terms, we have verified that the images have three channels for colored images. During analysis, empirically, we considered the hypothesis that black and white samples were in RGB format too.

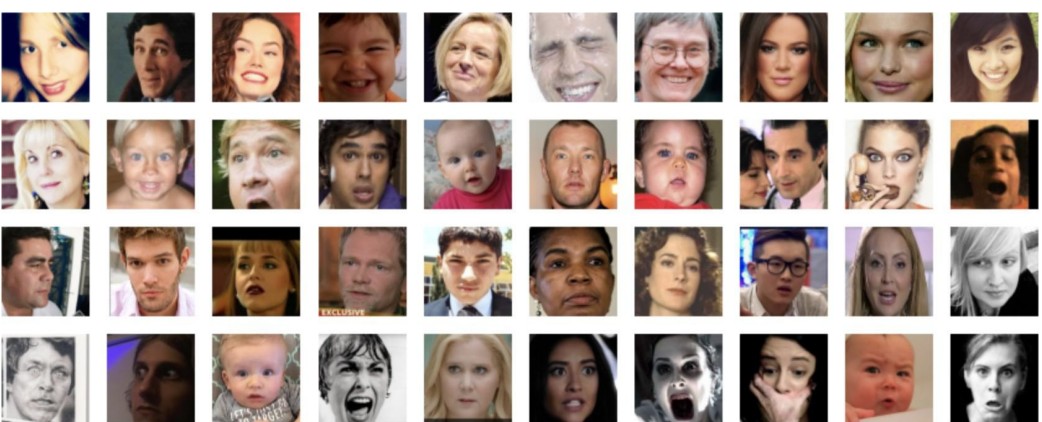

**Figure 5.** Samples from the AffectNet dataset.

Thus, we made sure to analyze the pixel values in the channel distribution, and we made the conclusion that for each of the pixels, we have the same value in the three channels. This aspect gives us the possibility to simply adapt FER-2013 and CK+48 to AffectNet samples using a channel conversion procedure and apply the brightness value, typically used for the pixel value in gray-scaled samples, for the three channels of the pixels of every image and transform them into black and white images in RGB format. The values of R, G, and B are 224, 196, and 184, respectively as shown in Figure 6. A problem encountered during the AffectNet analysis is given by the noise of the labels; indeed, this dataset is very poorly annotated, and such a problem surely impacts the overall accuracy.

The graph distribution of different classes in the AffectNet dataset is shown in Figure 7, where red bars denote values that are below the mean and green bars denote values that are above the mean, which is equal to 4690 for the training set and 490 for the validation set, respectively. Image extensions are totally saved in Google Drive, and each dataset uses RGB channels for colouring and comes in a variety of sizes (the total amount of data is around 2 GB).

We must establish a standard format to manage the datasets simultaneously. Finally, we have the final sections of interest in the fine-tuning phase and training on a few models that can be saved locally on the drive and used in an external application for real-time classification. More information about the modeling is provided in the following sections.

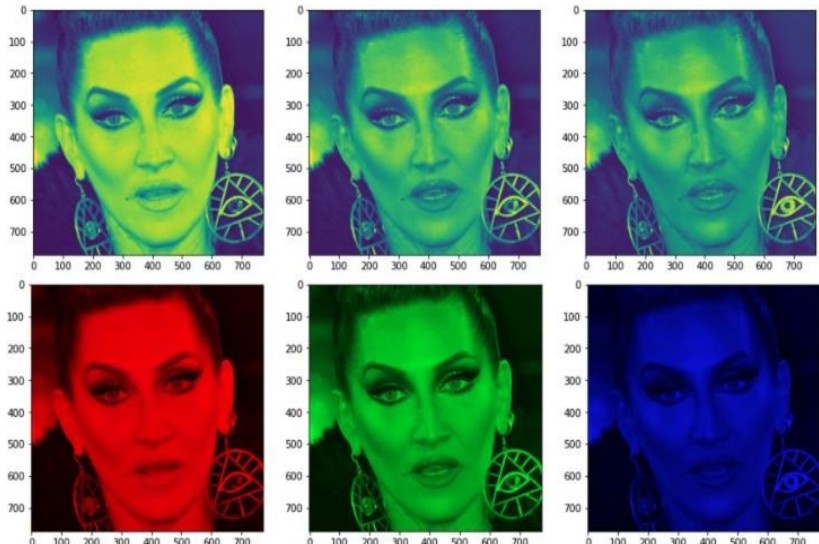

**Figure 6.** Pixel values for each channel are visualized using the zero matrices.

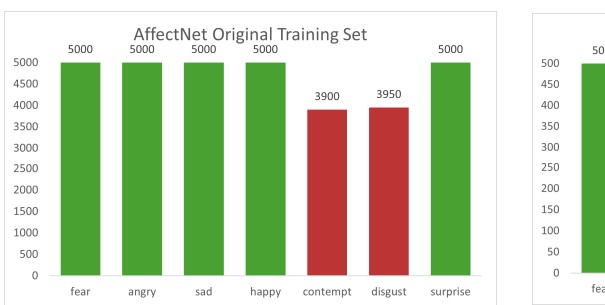
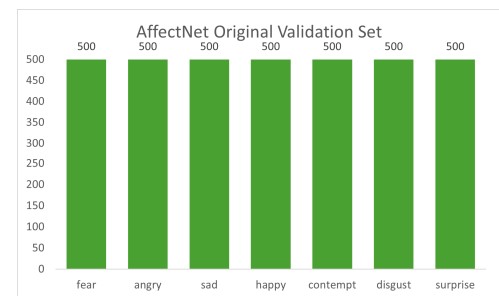

**Figure 7.** Distribution of classes in the AffectNet dataset; graph represents the mean values of green and red bars.

At the end of the data analysis for each data source, we merge them into the final dataset conventionally called AVFER (Aggregation for ViT on Facial Emotion Recognition). We have validation and testing sets balanced with the same number of samples for each class; meanwhile, the training set has a minimum amount of samples for every class of values but is not balanced. We will readjust the training set through data augmentation to reach sufficient samples for each class. We will then eliminate the excessive generated images and create a dataset with the same number of samples for each class of the problem domain. It is correct to say that the amount of data for contempt and disgust is very low, even after the integration with available open-source data; we can try to increase the variance of pixel matrices without using oversampling techniques and only using data augmentation, which increases the number of minor classes in the training set to obtain the same value of sample distribution and make the dataset balanced with generated images with similar features.

## 4. Preprocessing

This section discusses data manipulation in preparation for the *DataLoader* object from the *PyTorch* library. We transform the PNG images into JPG format as defined in Section 2; then, we convert the images into three channels and transform gray-scaled samples into RGB images. In conclusion, we performed data augmentation proportionate to each class's initial number of images.

*4.1. Data Manipulation*

AVFER has some samples from CK+ as PNG, and their images have one channel (gray-scaled type). We will avoid artificial coloring and reduce the side effects of the fooling image. So, we must convert it to RGB, modifying only the number of channels and adjusting pixel values. In the gray-scale samples, each pixel has 1 byte (8 bits equals a value from 0 to 255, corresponding to the image's brightness value). We must convert this pixel into three channels corresponding to 3 bytes equal to the red, green, and blue values during the transformation. In this conversion, each channel has the same value of brightness to maintain the gray tone of a colored pixel with no value perturbation. This operation uses the *OpenCVcvtColor* function, which obtains as parameters an image and a conversion type. For our objectives, we used the cv2.*COLOR BGR2RGB* to change the color and channels from a gray-scale representation to RGB. To ensure the robustness of the processing method, we have also performed color normalization using cv.normalize(). Finally, we compared the converted image with an original black and white sample of AffectNet to check if the distribution of the values follows the same roles.

*4.2. Data Augmentation*

Data augmentation has two final objectives:

1. Balancing the total amount of samples per class according to an arbitrary number chosen for the final dataset (20,000 samples per class).

2. Improving the total data of the training set and increasing the variability of patterns for the training phase. This feature is significant for transformer training because transformers need a huge amount of data to obtain good performances, even if we use fine-tuning.

Keras ImageDataGenerator class provides a quick and easy way to augment images. It provides various augmentation techniques defined as the parameters in the object declaration. It could be used during the preprocessing phase or even during the training phase during the epoch iterations. Due to the unbalanced class distribution, we decided to use it before the training phase. Expressly, we have set the following techniques:

1. **Zoom augmentation**: It either randomly zooms in on the image or zooms out of the image. ImageDataGenerator class takes in a float value for zooming in the *zoom range* argument. We used 0.6 because we want a high variance between generated images and to maintain a value smaller than 1 to zoom in on the image.

2. **Image rotation** is one of the widely used augmentation techniques that allows the model to become invariant to the orientation of the object. The value is between 0 and 360. We rotated images until 10 grades to adapt frontal images of FER-2013 and CK+48 on a similar face orientation to AffectNet faces and not cripple already rotated images.

3. **Random shifts**: It shifts the face of the image horizontally and vertically; our value is 0.2 for both types of shifting, and the manipulation is equal to moving pixels in different quantities until obtaining a maximum of 20% width and height images. Such manipulation can adapt images with some cropping on the face borders and always retains face parts such as eyes, noses, lips, and eyebrows that are typically the most relevant elements for facial emotion classification.

4. **Brightness augmentation**: It randomly changes the brightness of the image. It is also a helpful augmentation technique because faces will not be under perfect lighting conditions most of the time. Therefore, it becomes imperative to train our model on images under different lighting conditions. *ImageDataGenerator* defines a range to reduce or increase the brightness of an image; we have chosen a range of 0.2 to 0.6 to maintain a discernible pattern even in black and white images.

Generally, elements in the gray-scale mode are FER-2013 and CK+48 samples. Thus, we must convert them to RGB (three channels). In the gray-scale samples, each pixel has 1 byte (8 bits equals a value from 0 to 255, corresponding to the brightness value of the image as described in the gray-scale representation). During the transformation, we should convert the byte associated with one pixel in a three-channel representation corresponding

to 3 bytes equaling the red, green, and blue values. In this conversation, each channel has the same value of brightness to maintain the gray tone of the colored pixel without value perturbation.

We carried out data augmentation in proportion to the initial amount of samples per class. We divided the final amount that we wanted to reach (20,000 samples per class) by the initial amount.

The total number of copies for each sample depends on the total number of images for its class. We can calculate the number of copies *N* as follows:

$$C(i) = T \div N(i) + 1 \; \forall i \in classes.keys \tag{1}$$

where T is the total amount of samples that we want for each class, equal to 20,000; N(i) is the initial amount of samples for class i; and C(i) the final amount of data for class *i* after the augmentation phase. The result equals the number of augmented images we will generate. In other terms, we want to perform an oversampling approach to balance each class with at least 20,000 samples and undersampling on augmented images if the total number of samples overflows the class limit.

## 5. Models

In this section, we introduce the use of the Vision Transformer Single-Step Detector model for emotion classification and face cropping with adaptations.

### *5.1. Vision Transformers*

We exploit Vision Transformers by fine-tuning the pre-trained model on ImageNet to classify eight human emotions: anger, contempt, disgust, fear, happiness, neutral, sadness, and surprise as per Figure 8. Transformers are becoming the standard in NLP tasks. The core component in such a model is the attention mechanism, which can extract valuable features from the input with a standard query, key, and value structure, where the matrix multiplication between queries and keys pulls the similarity between them. Then, the softmax function applied to the result is multiplicated on the value to obtain our attention mechanism. Our transformer architecture is based on a stack of 11 encoders preceded by a hybrid patch embedding architecture. The improvement is made by considering the lack of an inductive bias problem. The Vision Transformers have much less image-specific inductive bias than CNNs.

### 5.1.1. Vision Transformer Base Structure

In the convolutional neural network, locality, two-dimensional neighborhood structure, and translation equivariance are baked into each layer throughout the entire model. MLP layers are local and translationally equivariant in the ViT configuration, while the self-attention layers are global. A two-dimensional neighborhood structure is used very sparingly. It is used at the beginning of the model by cutting the image into patches and at the fine-tuning time for adjusting the position embeddings for samples of different resolutions. Other than that, the position embeddings at initialization time carry no information about the patches' 2D positions, and all spatial relationships between the patches have to be learned from scratch. As an alternative to raw image patches, an input sequence is formed from feature maps of a CNN (LeCun et al., 1989). The patch embedding projection is applied to patches extracted from a CNN feature map in this hybrid model [29]. The embedding of the feature map is obtained by adding the position embedding and a linear projection of the feature map obtained from the CNN's output. If we want to obtain valuable results in terms of accuracy for our task, it is preferable to adopt a configuration of the training phase starting from a pre-trained model. The focus passes in a fine-tuning process where the initial classification domain (ImageNet) is adapted in an eight-dimensional classification task. We obtain this result by changing the last head layer with an eight-neuron linear layer (one for each class). Then, the model was fine-tuned with a cross-entropy Loss function

and the stochastic gradient descent algorithm. For the final prediction phase, to obtain a probability distribution over the eight classes, a softmax function is exploited.

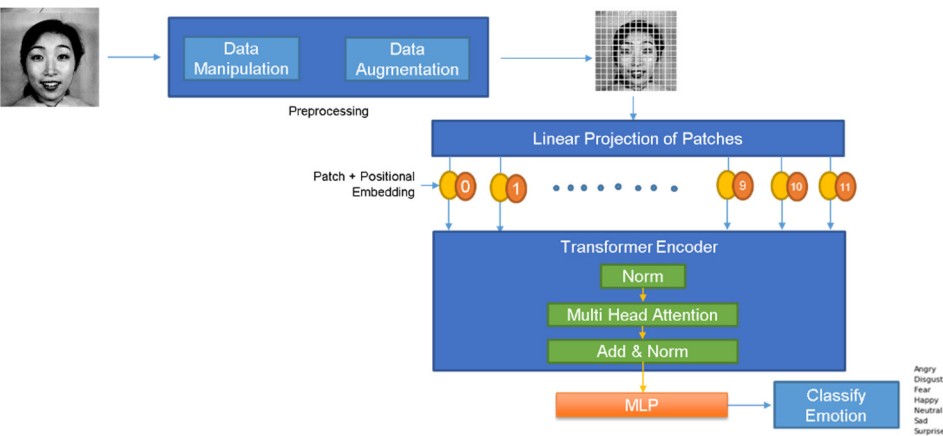

**Figure 8.** Architecture for emotion recognition using Vision Transformer.

5.1.2. Vision Transformer with Sharpness Minimizer

Sharpness-Aware Minimizer (SAM) uses are motivated by the connections between the geometry of the loss landscape of deep neural networks and their generalization ability. It is also used for transformers to smooth the loss landscape and simultaneously minimize loss values and loss curvature, thereby seeking parameters in neighborhoods having uniformly low loss values and generalizing, following a more linear curvature of the loss values. Therefore, we can modify the Vision Transformer described in the previous section by adapting the optimizer on the SAM. It has generalized the minimization of the loss value but increased the training time of the model [30].

$$\min_{\omega} \max_{\|\epsilon\|_2 \leq \rho} L_{\text{train}} (\omega + \epsilon)$$

The above equation [31] will be able to find the parameter w whose entire neighbors have low training loss $L_{\text{train}}$ by formulating a minimum and maximum objective.

The SAM optimization function can also tackle the "noisy label" problem presented by obtaining datasets with a high degree of robustness to label noise that indeed is a part of the problem when considering the AffectNet dataset as shown in Figure 9. Furthermore, according to final considerations, SAM works better with small datasets, mostly on Vision Transformers such as ViT-B/16. In conclusion, we have considered two assumptions: First, SAM incurs another round of forward and backward propagation to update neuron weights, which will lead to around 2x computational cost per update but result in better performances on small datasets. Second, we notice that the effect of the SAM diminishes as the training dataset becomes larger, but we cρannot have the non-augmented data and obtain a balanced dataset at the same time, due to the low number of contempt and disgust samples.

*5.2. Experiments*

The proposed model starts from pre-trained parameters given by the *Timm* library available in Python. It allows us to download a pre-trained model with specific transformer configurations based on the dimension of the last layer for fine-tuning. For each of the structures, it provides a random-weighted-based version without a pre-training phase. The model implementation uses PyTorch components as the main framework. During the preprocessing phase, we redefined the size of the images to adapt the dimensions of $224 \times 224$ on three different channels (corresponding to the RGB channels); we described data preprocessing more in detail in the previous section. Apart from the data augmentation phase, data follow some transformation during loading in the dataset object. This process

involves applying a random resize crop technique, which crops part of the image and resizes it into the input format to focus on aspects of the images and defines hidden patterns only in these, increasing feature classification in more detail. Then, we applied a random horizontal flip and an image-to-tensor transformation to adapt the pixel matrix in a PyTorch tensor object. Finally, we normalized the input data and prepared the samples for the training phase. The normalization phase applies a mean and a standard deviation to 0.5 for each channel. The best validation accuracy obtained from the set of epochs during the training phase defines the final model weight set. The fine-tuning phase adapts the model parameters to the FER task using stochastic gradient descent or Sharpness-Aware Minimizer adaptation with a cross-entropy loss function. The learning rate follows a scheduler that adjusts the initial value for every 10 epochs by multiplying it by 0.1; we implemented it, but we cannot follow it in practice because the training time available for our environment has a time limit of 24 h and we cannot reach more than 20 epochs in one cycle of training, so we dumped the weights of parameters and re-loaded them more times to continue the training in multiple sessions. Finally, we applied simple momentum of 0.9 to increase the speed of training and variable learning according to the optimizer chosen in the experiment. In conclusion, we used as the training environment Google Colab Pro with a P100 GPU and 16 GB of RAM available for a batch of 60 units. We carried out different experiments with various configurations of the Vision Transformer and compared it with the results of the state-of-the-art ResNet-18 model to verify the performance of Vision Transformers. The configurations of the ResNet-18 and ViT models are briefly described as follows:

1. **ResNet-18:** We trained it to compare our transformers with small, convolutional-based models to compare the trainability of our dataset on a model with a different structure and without the lack of inductive bias, which is typical for transformers trained on small datasets. We trained ResNet on a gradual learning rate decrement and momentum of 0.9 with SGD-based optimization on 25 total epochs.

2. **ViT-B/16/S:** It is the baseline model of ViT. It follows the basic Vision Transformer with 16 units on patch embedding and final linear layer adaptation for FER. This model uses SGD as an optimizer with a fixed learning rate equal to 0.001 and momentum of 0.9. We trained on 25 epochs.

3. **ViT-B/16/SG: This model has a similar** configuration to ViT-B/16/SGD but has a different learning rate, starting at 0.01. We used a gradual learning rate decrement. We trained on 25 epochs too.

4. **ViT-B/16/SAM:** It is equal to the baseline model ViT-B/16 with Sharpness-Aware Minimizer as the optimizer. We used a gradual learning rate decrement and a momentum of 0.9. We trained it on 25 epochs.

During the training phase, we saved histories of loss and accuracy for the training and validation sets. Every model version was trained on a hybrid and augmented training set formed by 161,000 images equally distributed in eight classes with AffectNet, FER-2013, and CK+48 samples and validated and tested only on an unused subset of AffectNet samples. We notice that noisy samples of AffectNet have a significant impact on performances. An example is the transformer adapted with attention-selective fusion reported by F. Ma, B. Sun, and S. Li in [30]; their proposed model obtains accuracy/robustness of between 56 and 62% on the complete version of AffectNet.

*5.3. Results of Experiments and Evaluation*

5.3.1. Analysis of Metrics for Various Models

The following Table 1 shows the principal classification metric results in the training phase. We tested models on 4000 different samples of AVFER without data augmentation and with disjoint relations with training and validation sets. The testing dataset is formed by 4000 samples equally distributed (500 samples per class).

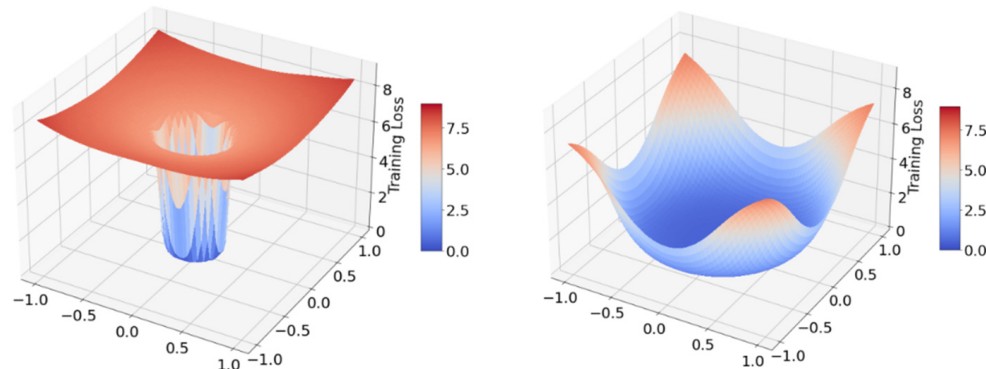

**Figure 9.** Cross-entropy loss landscape on ViT with SAM's application during the training on AVFER.

**Table 1.** Testing accuracy (with approximation to 8 and 7 classes), weighted average precision, recall, and F1-score on project models tested on AVFER.

| Metrics | Models Tested on the AVFER | | | |
|---|---|---|---|---|
| | ResNet-18 | ViT-B/16/S | ViT-B/16/SG | ViT-B/16/SAM |
| 8-Class Accuracy | 0.5005 | 0.5225 | 0.5242 | 0.5310 |
| 7-Class Accuracy | 0.5277 | 0.5489 | 0.5591 | 0.5694 |
| Weighted Avg. Precision | 0.5090 | 0.5485 | 0.5404 | 0.5470 |
| Weighted Avg. Recall | 0.6005 | 0.6225 | 0.6242 | 0.6310 |
| Weighted Avg. F1-Score | 0.4943 | 0.5184 | 0.6169 | 0.6220 |
| # Train. Params | 11.7 M | 86.5 M | 86.5 M | 86.5 M |
| # FLOPS | 1.8 G | 17.5 G | 17.5 G | 17.5 G |

Table 2 represents the average accuracy and loss incurred in ResNet-18, ViT-B/16/S, ViT-B/16/SG and ViT-B/16/SAM. Because of a lack of data for the contempt class, we evaluated the models on AffectNet considering only the seven less-augmented classes (original classes without contempt). Finally, for a more detailed evaluation, we have written the precision, recall, and F1-score distribution over classes and confusion matrix on the testing set for every tested model in the metric distribution section.

5.3.2. Training and Validation Evaluation

We tried some different configurations of SAM's use and the gradual learning rate on the ViT-B/16 configuration with the objective of finding the best configuration, avoiding overfitting or underfitting, and obtaining acceptable performances using a small dataset.

Figure 10 represents the training and validation accuracy change over time for the ViT-B/16 and ResNet-18 configurations that have been proposed. The main variances are caused by the transformers' steady learning rate, which results in a gradual increase in training accuracy along a regular curve, whereas the validation accuracy first experiences an increase before plateauing at about 55%, even for the ResNet-18. Hence, adopting the Vision Transformer on the gradual learning rate can improve the training accuracy curves, but performances in the validation accuracy remain similar.

The following Figure 11 represents how training and validation loss change for the proposed Vision Transformer configurations and compares them with the ResNet-18 behavior. According to the plots, ResNet-18 maintains a high training loss over time compared to ViT configurations. Due to the smoothing of the loss landscapes, the SAM-based ViT-B/16 in this work achieves a validation loss that is equivalent to ResNet-18, while the ViT-B/16/S and ViT-B/16/SG exhibit a comparatively high validation loss due to insufficient data.

**Table 2.** Accuracy and loss of various models on different configured parameters.

| No. of Epochs | Accuracy/Loss | ResNet-18 | ViT-B/16/S | ViT-B/16/SG | ViT-B/16/SAM |
|---|---|---|---|---|---|
| 5 | Training accuracy | 61.38 | 73.99 | 74.41 | 83.88 |
| | Validation Accuracy | 50.43 | 55.42 | 55.81 | 55.14 |
| | Training loss | 10.50 | 7.09 | 7.12 | 7.15 |
| | Validation loss | 14.70 | 12.61 | 13.35 | 12.82 |
| 10 | Training accuracy | 66.57 | 81.42 | 84.01 | 87.72 |
| | Validation Accuracy | 52.16 | 55.71 | 56.30 | 55.24 |
| | Training loss | 8.90 | 5.06 | 4.67 | 6.24 |
| | Validation loss | 14.90 | 14.33 | 15.05 | 13.39 |
| 15 | Training accuracy | 68.01 | 79.07 | 87.79 | 87.88 |
| | Validation Accuracy | 52.26 | 56.01 | 54.94 | 55.29 |
| | Training loss | 8.49 | 4.12 | 3.63 | 6.27 |
| | Validation loss | 14.80 | 19.47 | 17.58 | 13.16 |
| 20 | Training accuracy | 70.01 | 81.14 | 88.37 | 88.18 |
| | Validation Accuracy | 53.85 | 54.28 | 54.65 | 55.45 |
| | Training loss | 8.35 | 5.14 | 3.41 | 6.20 |
| | Validation loss | 14.20 | 15.03 | 18.76 | 13.08 |
| 25 | Training accuracy | 70.02 | 84.28 | 88.66 | 88.64 |
| | Validation Accuracy | 53.89 | 53.98 | 55.52 | 56.22 |
| | Training loss | 8.27 | 3.97 | 3.63 | 6.05 |
| | Validation loss | 1.39 | 19.93 | 17.87 | 12.93 |

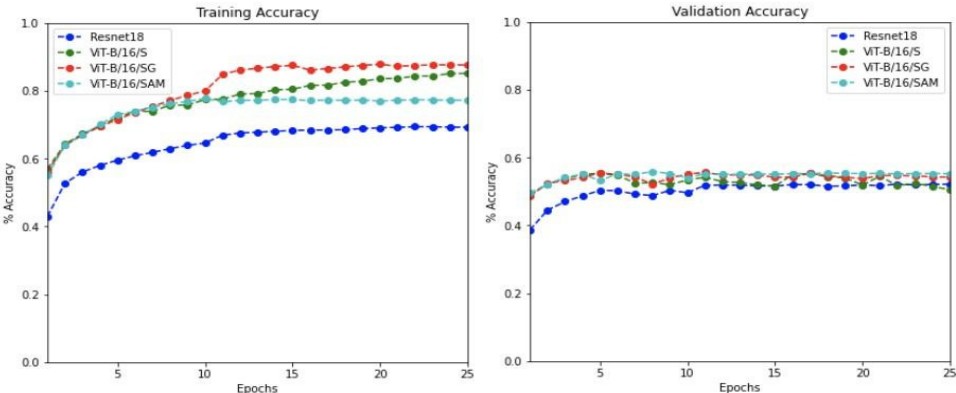

**Figure 10.** Training and validation accuracy for ResNet-18 vs. ViT-B/16 models.

5.3.3. Metric distribution on AVFER testing

Figure 12 represents the achieved values for Precision, Recall, F1-Score for each class of projected model on AVFER.

5.3.4. Confusion Matrices in AVFER Testing

According to Figure 13, we can see that performance is similar even for the ResNet configuration. Furthermore, the lack of contempt for original samples can affect performance in the correct classification of this class.

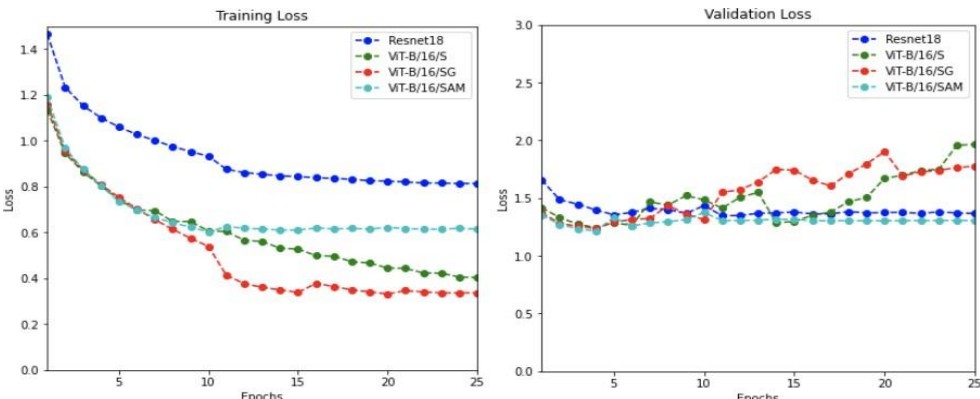

**Figure 11.** Training and validation loss for ResNet-18 vs. ViT-B/16 models.

| Classes | Precision | Recall | F1-Score |
|---|---|---|---|
| **Anger** | 0.4426 | 0.4780 | 0.4596 |
| **Contempt** | 0.5272 | 0.3100 | 0.3904 |
| **Disgust** | 0.5833 | 0.3360 | 0.4264 |
| **Fear** | 0.5846 | 0.5320 | 0.5571 |
| **Happy** | 0.5978 | 0.7460 | 0.6637 |
| **Neutral** | 0.3773 | 0.4520 | 0.4113 |
| **Sadness** | 0.4685 | 0.5800 | 0.5183 |
| **Surprise** | 0.4905 | 0.5700 | 0.5273 |

(**a**) ResNet-18

| Classes | Precision | Recall | F1-Score |
|---|---|---|---|
| **Anger** | 0.4264 | 0.6260 | 0.5073 |
| **Contempt** | 0.5348 | 0.3380 | 0.4142 |
| **Disgust** | 0.6644 | 0.3840 | 0.4867 |
| **Fear** | 0.5978 | 0.5380 | 0.5663 |
| **Happy** | 0.6144 | 0.7680 | 0.6827 |
| **Neutral** | 0.3850 | 0.6060 | 0.4709 |
| **Sadness** | 0.5464 | 0.5180 | 0.5318 |
| **Surprise** | 0.6185 | 0.4020 | 0.4873 |

(**b**) ViT-B/16/S

| Classes | Precision | Recall | F1-Score |
|---|---|---|---|
| **Anger** | 0.4746 | 0.5240 | 0.4981 |
| **Contempt** | 0.5957 | 0.2800 | 0.3810 |
| **Disgust** | 0.6224 | 0.3560 | 0.4529 |
| **Fear** | 0.5955 | 0.5800 | 0.5876 |
| **Happy** | 0.6121 | 0.7920 | 0.6905 |
| **Neutral** | 0.3736 | 0.5380 | 0.4410 |
| **Sadness** | 0.5374 | 0.5460 | 0.5417 |
| **Surprise** | 0.5115 | 0.5780 | 0.5427 |

(**c**) ViT-B/16/SG

| Classes | Precision | Recall | F1-Score |
|---|---|---|---|
| **Anger** | 0.4622 | 0.5380 | 0.4972 |
| **Contempt** | 0.6037 | 0.2620 | 0.3654 |
| **Disgust** | 0.6401 | 0.3700 | 0.4689 |
| **Fear** | 0.6302 | 0.5760 | 0.6019 |
| **Happy** | 0.5959 | 0.8080 | 0.6859 |
| **Neutral** | 0.4012 | 0.5320 | 0.4574 |
| **Sadness** | 0.5323 | 0.5940 | 0.5614 |
| **Surprise** | 0.5108 | 0.5680 | 0.5379 |

(**d**) ViT-B/16/SAM

**Figure 12.** Tables of precision, recall, and F1-score for each class.

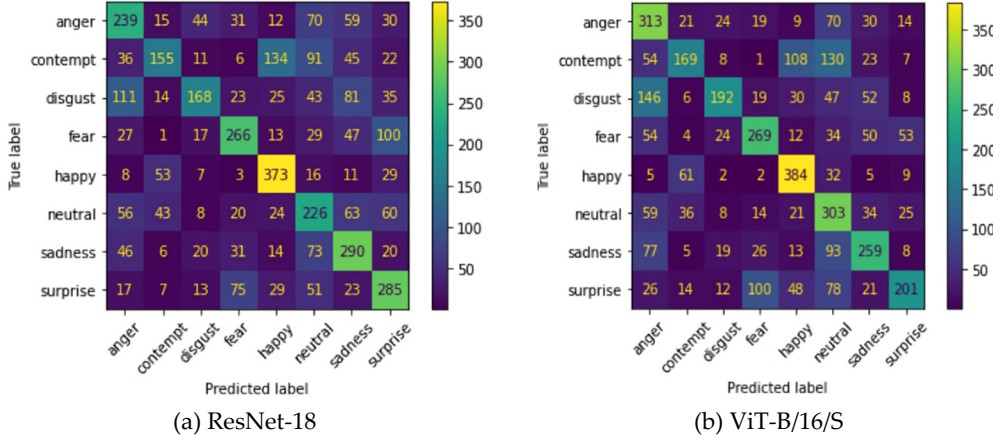

(a) ResNet-18

(b) ViT-B/16/S

**Figure 13.** *Cont.*

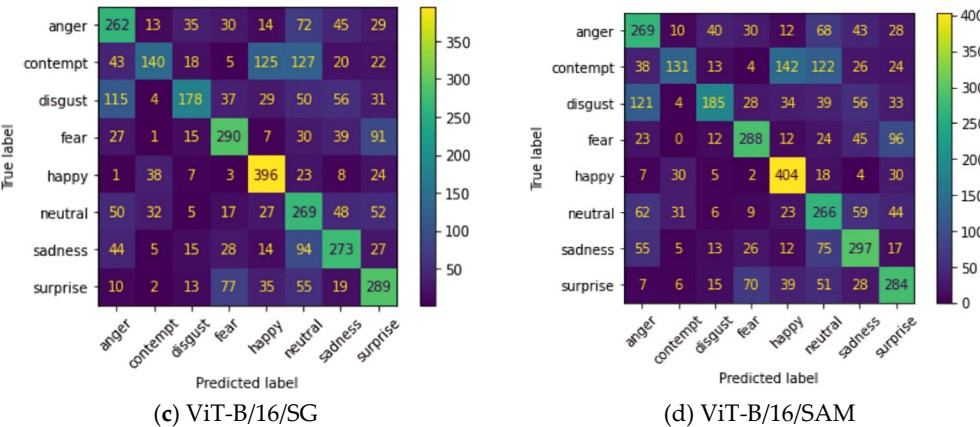

(**c**) ViT-B/16/SG    (**d**) ViT-B/16/SAM

**Figure 13.** Confusion matrix on AVFER testing.

## 6. Conclusions

In this study, we have applied fine-tuned ViT with a FER-based base configuration for image recognition and tested its robustness on AVFER. Firstly, we analyzed various datasets for facial emotion recognition by identifying the distribution ratio of emotion categories. After rectifying the uneven distribution of data, we have decided to merge three datasets, namely FER-2013, AffectNet, and CK+48 in the fraction of proportions. We have then split a set of sample data into training, validation, and testing to evaluate our model. Additionally, for the following model testing and evaluation, we contrasted the ViT-B/16 fine-tuned model with the similarly fine-tuned ResNet model. We have then decided to analyze the results by applying AVFER on the following models: ResNet-18, ViT-B/16/S, ViT-B/16/SG, and ViT-B/16/SAM, receiving an accuracy of 50.05%, 52.25%, 52.42%, and 53.10% and AUC of 0.843, 0.837, 0.801, and 0.589, respectively. Models with fewer FLOPS could speed up the entire prediction process in a scenario where speed is a limitation, such as in a real-time emotion recognition tool, which will be considered for future work. The carried-out research can help one choose the appropriate dataset and model for classifying emotions and will further assist in the area of research for emotion classification. This would ultimately help researchers identify various ViT models with the finest configuration. To conclude, based on our described model and achieved results, we believe that the proposed methodology will support additional research efforts in this area.

**Author Contributions:** A.C.; Conceptualization, Data curation, Methodoloy, Software, Validation, Visulization, Writing—original draft, C.B.; Conceptualization, Supervision, Project administration, Writing—review & editing, A.K.; Conceptualization, Data curation, Methodoloy, Software, Validation, Visualization, Writing—original draft, P.L.M.; Funding acquisition, Supervision, Writing—review & editing. All authors have read and agreed to the published version of the manuscript.

**Funding:** This research received no external funding.

**Data Availability Statement:** Data available in a publicly accessible repository that does not issue DOIs. Publicly available datasets were analyzed in this study. This data can be found on: [https://www.kaggle.com/datasets/msambare/fer2013] (accessed on 17 February 2022), [https://www.kaggle.com/shawon10/ckplus] (accessed on 20 April 2022) and [http://mohammadmahoor.com/affectnet/] (accessed on 2 May 2022).

**Conflicts of Interest:** The authors declare no conflict of interest.

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
