# Peer review of "ViTFER: Facial Emotion Recognition with Vision Transformers"

_asi, doi:10.3390/asi5040080_

Round 1
Reviewer 1 Report
Automatic emotion recognition based on facial expression is essential for improving machine intelligence that has a wide range of applications. In this paper, the authors examine the performance of a vision transformer for automatic emotion recognition. However, this study provides inadequate evidence to justify the claim that the proposed method improves the accuracy of automatic emotion recognition.
The reviewer strongly recommends a major revision on the following facts:
1. The introduction should be compelling to convince readers why this research is significant. The objective of the research must be remarked clearly, and the adopted research strategy should be introduced
2. The author should review the existing automatic FER methods, summarize the existing deep learning architectures method for FER and define the state-of-the-art of recognition rate.
3. What was the motivation behind selecting the ResNet18?
4. The reviewer recommends including a flowchart or algorithm to explain the proposed method.
5. The study fails to address how the findings relate to the existing works. The authors aim to demonstrate that the proposed emotion recognition system has improved accuracy, however, the data does not fully support this fact. The results of the proposed method should be compared with the relevant previous works to demonstrate the fact.
6. Conclusion should be synchronized with the objective of the work and give a clear scientific justification of the work.
7. The reviewer recommends applying color normalization to ensure the robustness of the proposed method. The RGB values are generated by some devices such as image devices or monitors. As RGB values are device dependent is not scientifically reasonable to use RGB values directly for object detection or image classification. Color normalization is a widely used preprocessing technique that reduces color variability caused by different devices by transforming the values into a common space.
8. The author should explain how the data augmentation methods were selected. Data augmentation is very useful while training a machine learning model which can be achieved by applying geometric transformation and generative transformation. However, it is necessary to select a set of augmentation methods suitable for facial emotion recognition. Vertical flip, Cropping, and Generative-based Transformation such as GAN can be considered for face data augmentation.
9. The authors carried out different experiments with different configurations to compare the performance. It is recommended to create a table containing the configurations.
10. The image quality of the figures should be improved. The explanation of Figure 2 should be clear and easy to interpret. For example, the order of emotions should be the same in both 2(a) and 2(b). The mean quantity for each graph should be mentioned.
Author Response
Reviewer 1
Comments and Suggestions for Authors
Automatic emotion recognition based on facial expression is essential for improving machine intelligence that has a wide range of applications. In this paper, the authors examine the performance of a vision transformer for automatic emotion recognition. However, this study provides inadequate evidence to justify the claim that the proposed method improves the accuracy of automatic emotion recognition.
The reviewer strongly recommends a major revision on the following facts:
- The introduction should be compelling to convince readers why this research is significant. The objective of the research must be remarked clearly, and the adopted research strategy should be introduced.
- We have added the objective, significance and strategy of our research in Introduction section.
- The author should review the existing automatic FER methods, summarize the existing deep learning architectures method for FER and define the state-of-the-art of recognition rate.
- We have reviewed the existing automatic FER methods in ‘Related works’ section and we have also summarized the deep learning methods for FER in 2.2 section, entitled ‘Emotion Recognition Methodologies Based on Deep Learning’.
- What was the motivation behind selecting the ResNet18?
- Our aim was to classify eight different types of emotions, in which ResNet18 is recommended. We also wanted to verify the efficiency of ViT model in accordance with CNN, so we have used ResNet18 as the convolutional neural network, as it is very useful and efficient in image classification.
- The reviewer recommends including a flowchart or algorithm to explain the proposed method.
- We have included the flowchart explaining the detailed algorithm and architecture of the proposed model in section 5.1 entitled, ‘Vision Transformers’.
- The study fails to address how the findings relate to the existing works. The authors aim to demonstrate that the proposed emotion recognition system has improved accuracy, however, the data does not fully support this fact. The results of the proposed method should be compared with the relevant previous works to demonstrate the fact.
- In order to satisfy the second comment, we have stated how our work is related to existing work in this field in 'Related Works' section. In this work, our aim was to experiment and explore various ViT models and compare it with one of the convolutional Neural Network model, which is ResNet-18. We do not want to undermine the performance of ResNet model, but our evaluations states the fact that the newly emerged ViT model has a better accuracy as shown in Table 1., furthermore we did not compare our proposed method with the relevant previous work as we have worked a completely new dataset namely AVFER, which is combination of three wellknown datasets namely, FER2013, CK+48 and AffectNet.
- Conclusion should be synchronized with the objective of the work and give a clear scientific justification of the work.
- Objective is synchronized with conclusion to add objective and justification of our work.
- The reviewer recommends applying color normalization to ensure the robustness of the proposed method. The RGB values are generated by some devices such as image devices or monitors. As RGB values are device dependent is not scientifically reasonable to use RGB values directly for object detection or image classification. Color normalization is a widely used preprocessing technique that reduces color variability caused by different devices by transforming the values into a common space.
- We have applied color normalization and mentioned about it in 4.1. Data manipulation section.
- The author should explain how the data augmentation methods were selected. Data augmentation is very useful while training a machine learning model which can be achieved by applying geometric transformation and generative transformation. However, it is necessary to select a set of augmentation methods suitable for facial emotion recognition. Vertical flip, Cropping, and Generative-based Transformation such as GAN can be considered for face data augmentation. – We have already applied cropping and zooming of facial expression, we will explain the same in paper.
- We applied the data augmentation methods before the training phase, due to the unbalanced class distribution. We have selected appropriate augmentation techniques based on the three datasets used in research. We have used zoom augmentation, image rotation, brightness augmentation and random shift that allows us to improve the data and increase the variability of patterns for training phase.
- The authors carried out different experiments with different configurations to compare the performance. It is recommended to create a table containing the configurations.
- Table 1 and 2 represents configuration for each model and their respective results, and training configurations are also mentioned in section 5.2 ‘Experiments’.
- The image quality of the figures should be improved. The explanation of Figure 2 should be clear and easy to interpret. For example, the order of emotions should be the same in both 2(a) and 2(b). The mean quantity for each graph should be mentioned
- We have reattached better quality images, replaced the old images and modified the explanation of all the figures appropriately.
Reviewer 2 Report
This paper uses multiple datasets to build a vision transformer-based expression recognition model. However, the work of this article is simply to use the vision transformer to recognize expressions, and there is basically no innovation.
Author Response
Reviewer 2
Comments and Suggestions for Authors
This paper uses multiple datasets to build a vision transformer-based expression recognition model. However, the work of this article is simply to use the vision transformer to recognize expressions, and there is basically no innovation.
- We have experimented on ViT transformer, with sharpness aware minimizer and we have done that on multiple datasets which are augmented to create a unique dataset that has not been experimented yet. The uniqueness of this dataset is that, it is the amalgamation of three well known datasets FER2013, CK+48 and AffectNet, and such experiment is yet not been carried out in the field of FER research.
Reviewer 3 Report
After reading this article, I suggest the authors complement the following data.
- Add a figure showing the structure of the vision transformer (on the 9th page). Understanding its base structure and inputs in the present manuscript is difficult.
- Besides, the required training parameters were not listed.
- I suggest the authors employ one or some equations to express the loss function. Just stating the use of a cross-entropy loss function in this article is insufficient.
- The scheme illustrating how to train the vision transformer wasn’t listed. Just mentioning the stochastic gradient descent or sharpness-aware minimizer adaptation is not enough.
- Although presenting that the proposed model outperforms other baselines using F1 scores is feasible, I suggest the authors calculate the AUC value (Area under the ROC curve) to show the scientific soundness of their works.
On the other hand, it is confusing that the authors named their proposed model ViT or ViTFER (in the title). Please revise this confusion.
Author Response
Reviewer 3
Comments and Suggestions for Authors
After reading this article, I suggest the authors complement the following data.
- Add a figure showing the structure of the vision transformer (on the 9th page). Understanding its base structure and inputs in the present manuscript is difficult.
- We have included the flowchart explaining the detailed algorithm and architecture of the proposed model in section 5.1 entitled, ‘Vision Transformers’.
- Besides, the required training parameters were not listed.
- Training parameters are mentioned in section 5.2 Experiments.
- I suggest the authors employ one or some equations to express the loss function. Just stating the use of a cross-entropy loss function in this article is insufficient.
- We have added an equation to express the loss function and also added cross entropy loss landscape graph for ViT with sharpness awareness minimizer in section 5.1.2. entitled ‘Vision Transformer with Sharpness Minimizer’.
- The scheme illustrating how to train the vision transformer wasn’t listed. Just mentioning the stochastic gradient descent or sharpness-aware minimizer adaptation is not enough. – mention parameters used in training of ViT.
- The method illustrating, how to train the vision transformer is mentioned in section 5.2 Experiments.
- Although presenting that the proposed model outperforms other baselines using F1 scores is feasible, I suggest the authors calculate the AUC value (Area under the ROC curve) to show the scientific soundness of their works.
- Taking the reviewers suggestion into consideration, we have added, AUC values of respective models in the conclusion section of our research paper.
On the other hand, it is confusing that the authors named their proposed model ViT or ViTFER (in the title). Please revise this confusion.
- ViTFER is an acronym created by merging two abbreviations ViT and FER which stand for Vision Transformer and Facial Emotion Recognition. Using this acronym we want to convey that we have applied the ViT model for Facial Emotion Recognition task.
Round 2
Reviewer 1 Report
Thank you for your deep and careful revision of the manuscript. The revised manuscript is easy to understand the importance of the work. The inclusion of Figure 8 is useful to understand the overview of the system. However, the reviewer recommends a minor revision on the following facts:
1. Section 5.3. should contain a discussion of the presented results. Especially the validation loss, which is very high as shown in Figure 11
2. The subtitles of Section 5.3 (Metrics Evaluation and Training Evaluation) can be modified, or the results can be sectioned more efficiently.
3. The image quality of Figure 12 and Figure 13 can be improved
Author Response
Thank you for your deep and careful revision of the manuscript. The revised manuscript is easy to understand the importance of the work. The inclusion of Figure 8 is useful to understand the overview of the system. However, the reviewer recommends a minor revision on the following facts:
- We truly appreciate you taking the time to provide us with your thoughtful feedback and suggestions. This has helped the manuscript to be improved and made simpler to interpret.
- Section 5.3. should contain a discussion of the presented results. Especially the validation loss, which is very high as shown in Figure 11
- In section 5.3, we have incorporated a more thorough explanation of the results as they relate to validation accuracy, validation loss, training accuracy, and training loss. We have also provided a brief justification for the high validation loss of ViT-16/B/S and ViT-16/B/SG in section 5.3.
- The subtitles of Section 5.3 (Metrics Evaluation and Training Evaluation) can be modified, or the results can be sectioned more efficiently.
- We greatly value the reviewers' suggestions, and we have revised the 5.3 title and subtitle as well as correctly sectioned the results.
- The image quality of Figure 12 and Figure 13 can be improved
- We have improved the images quality for figure 12. and 13.
Reviewer 2 Report
This paper mainly applies vision transformer to Facial Emotion Recognition. Basically, there is nothing innovative in this article, but fortunately, a more detailed test has been done on the data set.
Author Response
Thank you.
Reviewer 3 Report
No further questions. The review can end.
Author Response
Thank you.